Rapid eye movement-sleep is reduced in patients with acute uncomplicated diverticulitis—an observational study

Huang Chenxi 1 3 chenxihuang@msn.com
Alamili Mahdi 1
Nielsen Claus Henrik 2
Rosenberg Jacob 3
Gögenur Ismail 1
1 Department of Surgery, University of Copenhagen, Roskilde and Køge Hospitals , Køge , Denmark
2 Department of Infectious Diseases and Rheumatology, Institute for Inflammation Research, University Hospital of Copenhagen , Rigshospitalet, Copenhagen , Denmark
3 Department of Surgical Gastroenterology, University of Copenhagen, Herlev Hospital , Herlev , Denmark
Lee Yeong Yeh
Electronic publication date: 2015 Aug 13
Publication date: 2015
Volume: 3
Electronic Location ID: e1146
Received 2015 May 12; Accepted 2015 Jul 11
Copyright: © 2015 Huang et al.
Copyright year: 2015
Copyright holder: Huang et al.
License: This is an open access article distributed under the terms of the Creative Commons Attribution License, which permits unrestricted use, distribution, reproduction and adaptation in any medium and for any purpose provided that it is properly attributed. For attribution, the original author(s), title, publication source (PeerJ) and either DOI or URL of the article must be cited.
License URL: https://creativecommons.org/licenses/by/4.0/

Keywords: Rapid eye movement, Polysomnography, Diverticulitis, Cytokines, Infection, Inflammation, Sleep

Funding: The Lundbeck Foundation (Denmark) This work was supported by grants from The Lundbeck Foundation (Denmark). The funders had no role in study design, data collection and analysis, decision to publish, or preparation of the manuscript.

==============================
Introduction. Sleep disturbances are commonly found in patients in the postoperative period. Sleep disturbances may give rise to several complications including cardiopulmonary instability, transient cognitive dysfunction and prolonged convalescence. Many factors including host inflammatory responses are believed to cause postoperative sleep disturbances, as inflammatory responses can alter sleep architecture through cytokine-brain interactions. Our aim was to investigate alteration of sleep architecture during acute infection and its relationships to inflammation and clinical symptoms.

Materials & Methods. In this observational study, we included patients with acute uncomplicated diverticulitis as a model to investigate the isolated effects of inflammatory responses on sleep. Eleven patients completed the study. Patients were admitted and treated with antibiotics for two nights, during which study endpoints were measured by polysomnography recordings, self-reported discomfort scores and blood samples of cytokines. One month later, the patients, who now were in complete remission, were readmitted and the endpoints were re-measured (the baseline values).

Results. Total sleep time was reduced 4% and 7% the first (p = 0.006) and second (p = 0.014) nights of diverticulitis, compared to baseline, respectively. The rapid eye movement sleep was reduced 33% the first night (p = 0.016), compared to baseline. Moreover, plasma IL-6 levels were correlated to non-rapid eye movement sleep, rapid eye movement sleep and fatigue.

Conclusion. Total sleep time and rapid eye movement sleep were reduced during nights with active diverticulitis and correlated with markers of inflammation.

Introduction

Subjective and objective sleep disturbances are commonly found after major surgery (Gogenur et al., 2009; Kain & Caldwell-Andrews, 2003; Madsen, Rosenberg & Gogenur, 2013; Rosenberg, 2001). Several studies have demonstrated reduction or abolishment of slow-wave sleep (SWS) and rapid eye movement (REM) sleep during the first nights of sleep after surgery (Gogenur, Wildschiotz & Rosenberg, 2008; Hansen et al., 2013; Krenk, Jennum & Kehlet, 2012; Rosenberg et al., 1994). Sleep disturbances may give rise to several complications including cardiopulmonary instability, cardiac morbidity, transient cognitive dysfunction, inflammation, prolonged convalescence, fatigue, and hyperalgesia (Chouchou et al., 2014; Cremeans-Smith et al., 2006; Krenk, Rasmussen & Kehlet, 2010; Mullington et al., 2009; Rosenberg, 2001).

Many factors may influence postoperative sleep: discomfort including pain, use of drugs including opioids, nursing environments, psychological factors, effects of general anesthesia, and the surgical trauma itself (Chouchou et al., 2014; Rosenberg, 2001). Surgical trauma may lead to activations of metabolic, endocrine and inflammatory responses (Rosenberg-Adamsen et al., 1996), the surgical stress response. Sleep architecture can be altered by certain cytokines (IL-1β, TNF-α and IL-6) that are key mediators in the host inflammatory response (Pollmacher et al., 2000). This same universal host inflammatory response is also occurring during infections and after surgical trauma (Rosenberg-Adamsen et al., 1996). However, the isolated effect of this inflammatory response on sleep in a clinical population without surgery and general anesthesia has never before been investigated.

Our hypothesis was therefore that sleep architecture is altered during acute inflammation (diverticulitis), and the changes would correlate to level of circulating inflammatory cytokines. The aim of the present study was to report the sleep architecture in the course of acute uncomplicated colonic diverticulitis and investigate the relationship between acute inflammation, sleep disturbances, and subjective discomfort.

Materials & Methods

All patients were enrolled after giving written informed consent and the study was approved by the Ethics Committee of Capital Region of Denmark (ref: H-1-2012-155) and the Danish Data Protection Agency. The study was registered on www.clinicaltrials.gov (ref: NCT01840852).

We included patients between the age of 18 and 75 years, with The American Society of Anesthesiologists (ASA) Physical Status Classification I–III, and with a CT-scan confirming acute uncomplicated diverticulitis (Hansen and Stock stage I) (Klarenbeek et al., 2012). The onset of lower abdominal pain and fever had to be within 72 h prior to admission. The exclusion criteria were patients with diverticulitis requiring surgical intervention, complicated diverticular disease (fistula or abscess), onset of symptoms more than 3 days from inclusion, recent history (up to 3 months) of surgical intervention, myocardial infarction or arrhythmia, autoimmune diseases (Inflammatory bowel disease, lupus, multiple sclerosis), sleep disorders including obstructive sleep apnea, night shift professions, daily use of opioids, psychoactive drugs, or hypnotics, psychiatry disorders, daily consumption of more than 50 g of alcohol, diabetes mellitus, predicted poor compliance (language difficulties, etc.), pregnancy or nursing, urinary or fecal incontinence, severe kidney disease, or current cancer. Furthermore, sleep quality was screened with the Pittsburgh Sleep Quality Index (PSQI) to exclude any poor sleepers (PSQI > 5) (Buysse et al., 1989).

The study was composed of three nights of sleep monitoring in total, all from 2300 h (h) (lights out) to 0700 h (lights on) in a private room. After two successive nights of sleep monitoring, named study night 1 (S1) and study night 2 (S2), patients continued their treatment according to standard treatment guidelines for the department. Patients were re-admitted at 30 days after study night 1 to attend the last study night 3 (S3). In cases of relapse or failure of treatment during these periods, the patient was excluded. Patients were prohibited from intake of stimulating beverages or alcohol during the day of the study nights. Prior to lights out, light intensity in the room with all light sources off was measured with a luxmeter (Elma 1335 Luxmeter, Elma, Greve, Denmark), and only a level below 10 lux was acceptable. The blood samples were collected at 2230 h followed by administration of self-reported discomfort scales.

Polysomnography was performed with a portable recording unit (Embla titanium, Natus Medical Incorporated, San Carlos, California, USA). Continuous recording from 2300 h to 0700 h was made by using four-channel bipolar electroencephalography, EEG, (C4-M1, C3-M2, O2-M1, O1-M2), two-channel electrooculography, EOG, and two-channel electromyography (submental). Reusable 10 mm gold cup electrodes (Natus Medical Incorporated, San Carlos, California, USA) were applied for EEG, and single use self-adhesive electrodes were applied for EOG and EMG (Ambu® Neuroline 720, Ballerup, Copenhagen, Denmark). All electrodes were placed in accordance to The American Academy of Sleep Medicine (AASM) Manual for the Scoring of Sleep and Associated Events 2007 and the skin was prepared with light abrasion and alcohol swaps before electrode fixation. Sleep staging was performed in accordance to The AASM Manual for the Scoring of Sleep and Associated Events 2007. Staging was performed by one experienced, trained scorer blinded to study nights using the software Remlogic 3.2 (Natus Medical Incorporated, San Carlos, California, USA). Sleep recordings were staged in epochs of 30 s as either N1 (stage 1 sleep), N2 (stage 2 sleep), N3 (stage 3 sleep), REM (REM sleep) or W (awake). All other sleep parameters were calculated on the basis of these sleep stages. Total sleep time (TST) summed up the time spent across all sleep stages, while sleep onset (SO) summed up the time spent in wake stage before the first sleep stage. Wake after sleep onset (WASO) summed up the time spent awake after sleep onset and before final awakening. Lastly, number of awakening (NOA) showed the number awakening from any sleep stages.

Subjective measures concerning discomfort were administered 10 min before lights out, including 100-mm visual analogue scale (VAS) of pain at rest and pain during movements (0 mm = no pain and 100 mm = worst pain imaginable), a 9-point Karolinska Sleepiness Scale (KSS) ranging from extremely alert (= score 1) to fighting sleep (= score 9) (Akerstedt & Gillberg, 1990). Fatigue and malaise were measured in VAS of overall fatigue (0 mm = no sense of fatigue and 100 = worst fatigue imaginable) and general well-being (0 mm = extremely well and 100 = extreme malaise), respectively. Lastly, a 100-mm VAS of poor sleep quality (0 mm = best sleep and 100 mm = worst sleep imaginable) was also administered in the following morning to assess self-reported sleep quality.

At 2230 h blood samples were taken. The plasma was extracted and then stored at −80 degree Celsius until analysis. Tumor necrosis factor alpha (TNF-α), interleukin-1 beta (IL-1β), interleukin-2 (IL-2), interleukin-6 (IL-6), and interleukin-10 (IL-10) were measured in a Luminex 100 IS analyzer (Luminex Corporation, Austin, TX, USA) using appropriate multiplex antibody bead kits purchased from Invitrogen (Invitrogen Corporation, Carlsbad, CA, USA). Data were analyzed using StarStation version 2.0 software (Applied Cytometry Systems, Sheffield, UK). The lower levels of detection were as followed (in picogram/milliliter): TNF-α: 0.7, IL-1β: 0.4, IL-2: 2.8 IL-6: 0.2, IL-10: 0.4.

REM sleep time was the primary endpoint of this study, while the secondary endpoints included other sleep data, scores of self-reported discomfort and levels of circulating cytokines. We assumed that REM sleep duration was approximately 70 min for healthy subjects between the ages of 40 to 50 with a standard deviation of 30 min (Ohayon et al., 2004). A 25% reduction of REM sleep between study nights 1 and 3 was considered as of clinical importance. It was estimated that 11 patients were needed, when a two-sided significance level of 5% and a power of 0.90 were applied. Wilcoxon’s signed-rank test was used for all data (sleep data, self-reported discomfort scores and cytokine levels) as paired analyses between S1 and S3, and S2 and S3. Spearman’s correlation was used to analyze the correlations between changes in sleep stages (as per cent of total sleep time) and changes in cytokine levels from S1 to S3 and from S2 to S3. The correlations estimated were the differences, rather than the absolute values, between these measurements. The correlations were also calculated between changes in self-reported discomfort scores and changes in sleep stages, as well as changes in cytokine levels and changes in self-reported discomfort scores. All data were presented as medians and ranges or quartiles. IBM SPSS Statistics for Windows, Version 20.0 (Armonk, NY: IBM Corp.) was used to carry out the calculations. The level of statistical significance was set to 0.05.

Results

Thirteen patients were enrolled, 11 completed the study, and two were excluded, one due to recurrence of diverticulitis, and one due to technical failure of the PSG recordings. Characteristics of patients, diverticular disease and analgesics consumption are presented in Table 1. All 11 patients were treated with oral or intravenous antibiotics according to standard routines and oral paracetamol, if required also ibuprofen, during their admission.

Table 1 Characteristics of patients with acute diverticulitis and their analgesics consumption.

Subject characteristics			
Sex (male/female), n	5/6		
Age (years), median	52	(20–74)	
ASA (I/II/III), n	11/0/0		
Body Mass Index (BMI), median	28	(20–39)	
Pittsburgh Sleep Quality Index (PSQI), median	4	(2–5)	
Smokers, n	2		
Nicotine replacement therapy, n	0		
Disease characteristics			
Pain debut (hours), median	48	(24–72)	
Temperature at admission (°C), median	37.8	(36.9–39.1)	
Hansen and Stock stagea (0/I/II/III), n	0/11/0/0		
Cancer found with endoscopy 30 days after admission, n	0		
C-reactive protein level at admission (mg/l), median	94	(33–190)	
Analgesics administered			
Paracetamol, n	11		
Ibuprofen, n	2		
Morphine, n	0		
Notes.

Values are median and range. Stages I to III refer to increasing severity of diverticulitis. Stage I refers to acute uncomplicated diverticulitis, stage II refers to acute complicated diverticulitis and stage III refers to chronic recurrent diverticulitis. ASA, American Society of Anesthesiologists.

a Hansen and Stock staging of diverticulitis is a clinical classification that also accounts for asymptomatic diverticulosis (stage 0) (Klarenbeek et al., 2012).

All sleep data are presented in Table 2. TST values were significantly reduced from 391 min (median value) at S3 to 361 min and 374 min at S1 and S2, respectively. REM sleep was significantly diminished from 90 at S1 to 60 at S3. No changes in other sleep stages (N1, N2 or N3) were statistically significant. Wake time was, in contrast, significantly increased from 89 min at S1 to 119 min and 106 min, at S1 and S2, respectively. Wake time after sleep onset (WASO) was also extended from 50 min at S1 to 78 min and 86 min at S1 and S2, respectively. There was no statistical difference in number of awakenings between the study nights. Due to the risk of first night effect, post hoc paired analysis of TST, wake time and WASO between S1 and S2 were performed and showed no differences (W: p = 0.859, TST: p = 0.859, WASO: p = 0.533).

Table 2 Sleep parameters of patients with acute diverticulitis both during nights of the disease and during remission night.

Values are medians and interquartile ranges in the brackets. P-values for Wilcoxon’s signed rank test for study nights during diverticulitis compared to remission night (study night 3).

	Study night 1	Study night 2	Study night 3	
	Median	p-value	Median	p-value	Median	
N1 (min)	28 (24–40)	0.169	39 (32–51)	0.722	40 (32–52)	
N2 (min)	185 (122–199)	0.477	147 (138–161)	0.424	165 (146–191)	
N3 (min)	88 (73–102)	0.423	83 (50–103)	0.477	87 (74–101)	
Non REM (min)	285 (242–317)	0.091	284 (264–290)	0.062	293 (286–323)	
REM (min)	60 (42–84)	0.016	73 (47–92)	0.068	90 (76–112)	
W (min)	119 (92–184)	0.006	106 (94–155)	0.014	89 (61–109)	
Sleep onset (min)	39 (26–47)	0.248	26 (19–34)	0.594	29 (15–52)	
WASO (min)	78 (55–142)	0.005	86 (68–120)	0.003	50 (28–68)	
Awakening (number)	19 (17–27)	0.266	24 (23–29)	0.305	26 (19–31)	
TST (min)	361 (297–388)	0.006	374 (326–386)	0.014	391 (371–419)	
N1 of TST (%)	9.9 (6.0–12.5)	0.858	11.5 (8.5–14.1)	0.514	9.3 (7.7–14.5)	
N2 of TST (%)	47.4 (39.5–54.1)	0.398	45.3 (37.1–52.8)	0.449	42.8 (38.8–45.5)	
N3 of TST (%)	22.1 (19.5–27.5)	0.929	23.8 (18.5–26.8)	0.610	21.9 (19.5–26.1)	
NREM of TST (%)	82.5 (76.5–87.5)	0.032	80.6 (75.5–86.5)	0.139	75.5 (72.5–78.0)	
REM of TST (%)	17.5 (12.5–23.5)	0.036	19.4 (13.5–24.5)	0.139	24.5 (22.0–27.6)	
Notes.

Nx stage x sleep

REM rapid eye movement sleep

W wake time

WASO wake after sleep onset

TST total sleep time

Cytokine levels are depicted in Fig. 1. Vast majority of IL-1β and IL-2 measurements were found below the detection thresholds, therefore they were excluded from further analysis. IL-6 levels were significantly decreased from 6.3 picogram/milliliter (median) at S1 to 5.1 picogram / milliliter at S2 (p = 0.008). IL-6 levels was reduced to 0.2 (p = 0.051) at S3, but without statistical significance. IL-10 (p = 0.128 for both nights versus S3) and TNF-α (p = 0.103 for S1, p = 0.314 for S2) levels were decreased over study nights, however, without statistical significance.

Figure 1 Plasma concentrations of cytokines in patients with acute diverticulitis both during nights of the disease and during remission night.

Significant differences between the nights of the disease and remission night were calculated using Wilcoxon’s signed rank test and are indicated with ∗p < 0.05, ∗∗p < 0.01. The horizontal broken lines indicate the detection thresholds for the individual cytokines. The whiskers depict interquartile range. The cytokines measured include: IL-10, interleukin 10; IL-6, interleukin 6; TNF-α, tumor necrosis factor alpha.

Subjective discomfort scores are depicted in Fig. 2. Malaise (p = 0.003, p = 0.005), pain during movements (p = 0.005, p = 0.005), resting pain (p = 0.003, p = 0.008) and poor sleep quality (p = 0.003, p = 0.041) were significantly diminished at S1 and S2 compared with S3. The fatigue was also diminished, however only with statistical significance from S1 to S3 (p = 0.006), but not from S2 to S3 (p = 0.374). Lastly, no significant differences were found for sleepiness for both nights (p = 0.892, p = 0.619, respectively).

Figure 2 Subjective discomfort scores in patients with acute diverticulitis both during nights of the disease and during remission night.

Significant differences between the nights of the disease and remission night were calculated using Wilcoxon’s signed rank test and are indicated with ∗p < 0.05, ∗∗p < 0.01. The whiskers depict interquartile range. Apart from sleepiness, which was measured with KSS, Karolinska Sleepiness Scale; remaining discomfort domains were measured with VAS, visual analogue scale.

Since the vast majority of IL-1β and IL-2 measurements were below the detection limits, they were excluded from correlation analyses. The correlations between cytokine levels and sleep stages are stated in Table 3. We found significant associations in changes in IL-6 levels and changes in sleep stages as percentage of TST from both S1 and S2 to S3. There were strong correlations between IL-6 and REM (S1: r = − 0.68, p = 0.02 and S2: r = − 0.67, p = 0.03) and NREM (S1: r = 0.68, p = 0.02 and S2: r = 0.67, p = 0.03). In addition, a moderate correlation between IL-6 and N2 (S1: r = 0.60, p = 0.05 and S2: r = 0.63, p = 0.04) was found. We did not find any significant correlations between changes in self-reported discomfort and changes in sleep stages; however, we did find a positive correlation between changes in fatigue and IL-6 levels from S1 to S3 (r = 0.61, p = 0.05). No other significant associations were found between changes in discomfort and changes in cytokine levels.

Table 3 Correlation cross table for changes in sleep stages and cytokine levels.

The changes for study night 1 were defined as values of study night 1 minus study night 3 (remission night), and for study night 2 it was values of study night 2 minus study night 3. Significant correlations are presented in bold. Values are correlation coefficients and the corresponding p-values.

		↑ Non REM	↑ REM	↑ N1	↑ N2	↑ N3	
		Night 1	Night 2	Night 1	Night 2	Night 1	Night 2	Night 1	Night 2	Night 1	Night 2	
↑ IL-6	Correlation	0.68	0.67	−0.68	−0.67	−0.18	−0.48	0.60	0.63	−0.24	−0.18	
p-value	0.02	0.03	0.02	0.03	0.60	0.14	0.05	0.04	0.47	0.61	
↑ IL-10	Correlation	0.04	0.36	−0.04	−0.36	0.24	0.17	−0.21	0.12	0.22	−0.03	
p-value	0.91	0.28	0.91	0.28	0.48	0.62	0.53	0.72	0.51	0.93	
↑ TNF-α	Correlation	−0.26	0.13	0.26	−0.13	−0.04	0.40	−0.18	0.03	0.30	−0.08	
p-value	0.43	0.71	0.43	0.71	0.90	0.22	0.59	0.93	0.38	0.83	
Notes.

REM rapid eye movement sleep

Nx stage x sleep

IL-x interleukinx

TNF tumor necrosis factor

Discussion

We used acute uncomplicated diverticulitis as a model to investigate the effects of inflammatory responses on sleep and found that TST was consistently reduced at both study nights with diverticulitis (S1 and S2) due to increased W and WASO, and that REM sleep was decreased at S1. We also found that subjective discomfort scores were increased except sleepiness at both diverticulitis nights. Lastly, we found a significant increase in IL-6 levels during the study nights during which the patients had diverticulitis, and this increase was correlated to changes in NREM sleep (including N2 sleep), REM sleep, and fatigue scores.

Suppression of REM sleep and promotion of WASO and NREM sleep found in this study are similar to findings observed in clinical polysomnography studies of patients with infections such as early stage HIV infection and acute rhinovirus infection (Opp, 2009). In addition, sleep studies using endotoxin have been performed in healthy humans (Pollmacher et al., 2000), showing increased wake time, NREM sleep and decreased REM sleep after endotoxin administration (Mullington et al., 2000; Trachsel et al., 1994). In addition, the endotoxin-induced IL-6 elevation was positively correlated with REM latency, N2 sleep and wake time, and negatively correlated with REM sleep (Pollmacher et al., 1993). The reducing effect of REM sleep on IL-6 levels has also been demonstrated in another study (Spath-Schwalbe et al., 1998). One cross-sectional study has shown that levels of circulating IL-6 within normal ranges are significantly correlated to sleep architecture (increase of REM latency and WASO) when demographical variations were controlled (Hong et al., 2005). These findings are consistent with the associations we found in the present study, and further support the growing evidences of the sleep regulating roles of pro-inflammatory cytokines (Bryant, Trinder & Curtis, 2004). So far, evidence suggests that IL-6 can act directly on the central nervous system (Imeri & Opp, 2009), including activation of the hypothalamic-pituitary-adrenal axis (HPA-axis), to exert its effect on sleep regulation (Gomez-Gonzalez et al., 2012). Furthermore, studies have shown that inhibition of the IL-6 receptor by tocilizumab can alleviate sleep disturbances and fatigue in patients with rheumatoid arthritis (Rohleder, Aringer & Boentert, 2012; Thomas et al., 2011).

The elevated levels of plasma IL-6 found in the current study are coherent with findings in recent clinical studies investigating IL-6 levels during the course of acute diverticulitis and other abdominal infections (Elsing, Ernst & Stremmel, 2012; Rivera-Chavez et al., 2003). However, IL-1β and IL-2 levels were not elevated or detectable at all. This discrepancy has previously been observed in patients with acute pancreatitis and appendicitis (Brivet, Emilie & Galanaud, 1999; Rivera-Chavez et al., 2003). It is possible in the current study that levels of IL-1β and IL-2 may have peaked before the blood sampling, and the cytokines were already bound by high level of endogenous soluble receptors (Brivet, Emilie & Galanaud, 1999).

Compared to postoperative sleep disturbances (Rosenberg, 2001), REM sleep suppression found in our study was less prominent (33%). SWS and REM sleep are most often absent (100% reduction) during the first nights after major non-cardiac surgery (Rosenberg, 2001). In the postoperative period, pain and disturbing hospital environment (noise and staffs) are likely to cause awakenings during the night (Rosenberg, 2001), while opioids for pain relief can reduce SWS and REM sleep (Shaw et al., 2005). In recent years, more major surgeries have been performed with fast track regimes, in which perioperative care has been optimized with multimodal opioid-sparing analgesia and early hospital discharge (home environment) (Kehlet, 2008). Despite these efforts, REM sleep in the first postoperative night is still greatly reduced (62–92%) (Dette et al., 2013; Hansen et al., 2013; Krenk, Jennum & Kehlet, 2012). One randomized trial also demonstrated that even when opioids were avoided and pain was well managed, REM sleep was still abolished after major surgery (Cronin et al., 2001). In contrast, minor or less invasive surgery, such as laparoscopic surgery, only causes transient SWS reduction (46%), leaving REM sleep unaffected (Rosenberg, 2001). These findings may indicate that the magnitude of the surgical trauma, including the inflammatory response, plays a greater role in the postoperative sleep disturbances, compared to pain and opioid usage. Laparoscopic procedures cause less trauma and inflammatory response, manifested by CRP, IL-6, and TNF-α compared to major surgery (Akhtar et al., 1998; Braga et al., 2002), and therefore only causing a transient shift from SWS sleep to shallow NREM sleep.

Our study has some limitations. Patients were only mildly affected by their disease, and the magnitude of inflammation was therefore low, yet enough eliciting elevation of cytokines, C-reactive protein and body temperature. In our institution, these cases were treated with antibiotics and paracetamol, and to our knowledge, these drugs will not cause any (REM) sleep alteration. Due to the nature of the acute diverticulitis, the investigated periods (S1 and S2) of host inflammatory response were most likely not exactly synchronized amongst the patients. We did, however, only include patients with pain onset less than 72 h prior to the admission and inflammation in the diverticula proven by CT-imaging and elevated C-reactive protein. Therefore, the inflammatory response was present during the first study night, but the magnitude of this response could still vary between the patients, and so could their recovery rate due to the disease states and age differences (Ottinger et al., 2014). These uncertainties were attempted to be dealt with using paired observations, where each patient was its own control, thus allowing for differences in baseline values between the patients.

First night effect refers to reduction of TST and increase of wake time and WASO due to exposure to unfamiliar sleep environment (Toussaint et al., 1995). We did not find any significant changes in these sleep parameters between S1 and S2.

Two of the included patients were active smokers, who due to hospitalization (smoking prohibition) might have smoked lesser during the study sessions. None of the included patients reported changes in their smoking status during the entire course of the study. Epidemiological studies suggested that cigarette smoking was associated with difficulty in sleep initiation (increased sleep latency) and maintenance (Wetter & Young, 1994; Zhang et al., 2006). Therefore it is counter-intuitive that the sleep latency and awakenings after sleep onset were reducing from S1 to S3 for both patients in the current study. In addition, our main finding, approximately 30% REM-sleep reduction, cannot be explained by cigarette smoking (Zhang et al., 2006). The nature of our findings was, however, in line with an immune response mediated sleep disturbance (Spath-Schwalbe et al., 1998).

Conclusion

In conclusion, we found substantial sleep disturbances and subjective complaints in relation to acute uncomplicated diverticulitis. Inflammatory parameters correlated to sleep stages and subjective complaints.

Supplemental Information

Supplemental Information 1 Raw Dataset

Some sleep parameters, such as Total Sleep Time, are not included in the Raw dataset, because they are calculated from the raw sleep data. E.g. Total Sleep Time = sum of time spent in all four stages of sleep.

Click here for additional data file.

Additional Information and Declarations

Competing Interests

Author Contributions

Human Ethics

The authors declare there are no competing interests.

Chenxi Huang conceived and designed the experiments, performed the experiments, analyzed the data, contributed reagents/materials/analysis tools, wrote the paper, prepared figures and/or tables, reviewed drafts of the paper.

Mahdi Alamili and Claus Henrik Nielsen analyzed the data, contributed reagents/materials/analysis tools, reviewed drafts of the paper.

Jacob Rosenberg and Ismail Gögenur conceived and designed the experiments, analyzed the data, contributed reagents/materials/analysis tools, reviewed drafts of the paper.

The following information was supplied relating to ethical approvals (i.e., approving body and any reference numbers):

The Ethics Committee of Capital Region of Denmark, approval number: H-1-2012-155.

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
