# Peer review of "Rapid eye movement-sleep is reduced in patients with acute uncomplicated diverticulitis—an observational study"

_PeerJ, doi:10.7717/peerj.1146_

## Round 0.1 · original submission · Minor Revisions

Overall, reviewers have favorable comments on the study, and authors are advised to revise to meet their minor concerns. Please do check your abstract on the number of participants recruited and analyzed.

Reviewer 1 ·

Basic reporting

Good.

Experimental design

No Comments.

Validity of the findings

No Comments.

Additional comments

Interesting findings of an inflammatory model affecting sleep.

·

Basic reporting

No comments

Experimental design

No comments

Validity of the findings

No comments

Additional comments

The relevance of the statement " Since post-operative sleep............surgical patients " in lines 61- 63 is not understandable after stating the hypothesis.Moreover, what the superscripts used in these lines indicate is not clear. Is it references? The relevance of this statement here needs explanation.

The statement in lines 149 and 150 " TST values were significantly reduced..........increased" is not clearly understandable. The statement here should be clearly expressed as per data stated in the table.

Similarly the statement in lines 156, 157, 158 and 159 " IL-6 levels..........significance" is not clearly understandable. The statement here should be clearly expressed as per data stated in the figure.

In table1 , (line 361) the total number of male and female, analgesic intake are stated under the column of median. The table needs to be revised.

·

Basic reporting

No comments

Experimental design

No comments

Validity of the findings

No comments

Additional comments

1. Were any of the subjects active smokers or on smoking-cessation therapy (e.g. nicotine patches)? As the act of smoking may be stimulatory in nature and affect your results (or alternatively sudden cessation of smoking may affect the patient's sleep cycle).
2. Line 60/61: "complications that complicate..." suggest slightly change sentence to make it sound better.

---

## Round 0.2 · accepted · Accept

We agree that the revised version is acceptable for publication in its current form. Congratulations on the work.

·

Basic reporting

No comments

Experimental design

No comments

Validity of the findings

No comments

Additional comments

Comments regarding the possibility of smoking interfering with sleep cycles have been addressed by the author in the discussion.